# Preference Drivers for Blackberry Nectar (*Rubus* spp., Rosaceae) with Different Sweeteners

**DOI:** 10.3390/foods12030549

**Published:** 2023-01-26

**Authors:** Helena Maria André Bolini, Rafael Sousa Lima, Raquel Linhares de Freitas, Alessandra Cazelatto de Medeiros

**Affiliations:** Department of Food Engineering and Technology, School of Food Engineering, University of Campinas UNICAMP, Campinas 13083-970, SP, Brazil

**Keywords:** time-intensity analysis, consumer acceptance, sensory analysis, blackberry nectar, sweeteners, partial least square regression analysis

## Abstract

This study determined the dynamic sensory profile and consumer acceptance of blackberry nectar with different sweeteners. The ideal scale was used to determine the ideal sweetness of the sucrose and the magnitude estimation method for the equivalent sweetness of the sweeteners. The sensory profile was determined by time-intensity analyses with trained panelists. This study determined the dynamic sensory profile and consumer acceptance of blackberry nectar with different sweeteners. First, to determine the concentration of sucrose to promote optimal sweetness in blackberry nectar, a study was carried out by consumers, who used an unstructured 9 cm “Ideal Scale”, ranging from the extreme left as “extremely less sweet than ideal” to the extreme right as “extremely sweet than ideal”, with the center of the scale being the ideal sweetness point. Then, the magnitude estimation method was applied to determine the concentration of each sweetener studied in order to obtain the same sensation of ideal sweetness in the blackberry nectar. The sensory profile of blackberry nectar in the same equi-sweetness was determined by time-intensity analysis with trained assessors and CATA (Check-All-that-Apply) with consumers. According to our results and the opinion of the involved consumers, the optimal sucrose concentration in blackberry nectar was 9.3%, and the sweetener concentrations equivalent to sucrose were 0.015% of sucralose, 0.052% of aspartame and 0.09% of stevia with different rebaudioside A concentrations. Time intensity and overall liking data were statistically analyzed by partial least squares regression (PLSR), thus generating the temporal preference drivers for blackberry nectar. The results showed that the sucralose and tasteva sweeteners have a temporal profile closer to sucrose, being characterized by a lower intensity and duration of sweet and bitter taste, with a positive impact on consumer acceptance. Concomitant results were found by the CATA analysis, indicating that the attributes of blackberry aroma, blackberry flavor, sweet taste, and brightness also have a positive impact and stand out in the samples with sucrose, sucralose, and tasteva. The samples sweetened with stevia were characterized by a greater intensity of bitter taste and the presence of a sweet and bitter aftertaste, with a negative impact on acceptance. The different rebaudioside A concentrations in stevia (78%, 92%, and 97%) did not interfere with consumer acceptance.

## 1. Introduction

The World Health Organization, concerned about the risk factors associated with excessive sugar consumption, has updated consumption recommendations for children and adults to less than 10% of total daily energy intake, with even more beneficial effects at intakes less than 5%. These recommendations include all added sugars, as well as the naturally occurring sugars in honey, fruit nectars, syrups and fruit nectar concentrates [1].

Sugary drinks represent an important source of added sugar in the diet, and therefore many products are being developed to reduce sugar intake; however, reducing the amount of sugar can alter the sensory properties and the consumer’s hedonic response [2]. One of the alternatives to replace sucrose are sweeteners, allowing the sensation of the sweet taste that sucrose provides, even when using in smaller amounts and without calories. Despite the various sweeteners allowed by legislation, each one has specific characteristics, such as the intensity of the sweet taste, or the presence or absence of residual taste, which can influence the sensory properties of the product [3].

The current consumer market emphasizes taste, variety, nutritional benefits, and fresh foods, as well as behavioral changes, which have generated the development of food products with functional or nutritional ingredients [4]. Thus, beverage industry producers are oriented to respond to the emerging demand for natural beverages as a result of the therapeutic benefits they provide, in addition to the basic nutritional properties. Natural fruit and vegetable drinks are tasty, nutritious, and rich in vitamins, minerals, phytonutrients, and more precisely bioactive compounds [5]. Thus, it is important to select the type of fruit and define the necessary ingredients to elaborate a tasty product, without added sugars and with health benefits. In this perspective and in view of the great diversity of fruits, blackberries are highlighted due to their composition that is rich in phenolic acids, anthocyanins, procyanidins and flavonoids, in addition to the pleasant flavor and characteristics of juicy fruit that are in the form of a group of small berries growing on shrubs or vines [6]. Consuming blackberries gives the body several nutritional and health-promoting benefits. The compound with antioxidant activity contained by blackberries helps the human organism to fight against infectious diseases [5]. The sweeteners are applied, each time more, to replace the sucrose, because confer sweetness without calories [3].

The sweetener power of stevia and its derived of more percentage of rebaudioside A what is the principal sweet compounds in the plant (stevia rebaudioside A 78%, stevia rebaudioside A 92%, stevia rebaudioside A 97%, and tasteva) the sucralose and aspartame, currently, are the sweeteners with more application in foods, because the studies show that these sweeteners are considered sensorially suitable to replace sucrose by the consumer [3]. However, the scientific literature evidence presents results proving that the acceptance, sweetness power, and characteristics of these sweeteners depend highly of the product where they are. For example, the dispersion medium, acidity, viscosity, and other physical-chemistry characteristics [2].

The black mulberry presents the high adaptation to different climatic conditions and has a low production cost; however, the fresh fruit has a limited market due to its fragility and perishability, requiring a suitable storage structure. The ideal temperature is between 2 and 5 °C in order to inhibit enzymatic activity and microbiological development and preserve attributes such as flavor, color, and texture, which reduces its *in natura* consumption [7]. An alternative to this limitation is the industrialization of blackberries, which can be sold as fruit pulp for later use in nectars or frozen desserts [8].

In view of the above, the objective of this work was to determine the concentration of sweeteners needed to promote a sweetness equivalent to the ideal sweetness with sucrose and to determine the dynamic sensory profile of blackberry nectar with the use of different sweeteners through a time-intensity analysis, as well as its acceptance and characterization by consumers.

## 2. Material and Methods

### 2.1. Materials

The blackberry nectar samples were prepared at the Laboratory of Sensory Science and Consumer Studies of the Faculty of Food Engineering at UNICAMP. The nectar was prepared using pasteurized blackberry pulp (Ricaeli ^®®^, Cabreuva, SP, Brazil) and mineral water in the proportion of 1:2 (m/m), then homogenized in a blender for three minutes. The samples were subsequently sweetened with sucrose (União^®®^, Sao Paulo, SP, Brazil) and six different sweeteners, homogenized for another minute. The sweeteners used were: sucrose (União ^®®^, Sao Paulo, Brazil, Aspartame (SweetMix ^®®^, Campinas, SP, Brazil), stevia with 78% of rebaudioside A (Steviasoul ^®®^, Maringá, PR, Brazil), stevia with 92% of rebaudioside A (Clariant ^®®^ Steviasoul ^®®^, Maringá, PR, Brazil), stevia with 97% of rebaudioside A (Clariant ^®®^ Steviasoul ^®®^, Maringá, PR, Brazil), and tasteva ^®®^ (Mastersense^®®^, Jundiai, SP, Brazil).

The samples were prepared in the same way for all the methods.

### 2.2. Physicochemical Characterization

The blackberry pulp used in the nectar preparation was characterized through physicochemical analyses carried out at the central laboratory of the Department of Food Science and Nutrition at FEA/UNICAMP.

#### 2.2.1. Titratable Acidity

The titratable acidity was performed according to the total titratable acidity method of the Official Methods of Analysis of AOAC INTERNATIONAL [9]. An aliquot of 1 g of blackberry pulp was completed with 100 mL of distilled water and titrated with 0.1542 N NaOH solution until the turning point (pH 8.2), which was observed using a pH meter. The result was expressed in citric acid% by Equation (1):(1)$$\mathrm{Citric}\;\mathrm{acid}\;\%=\frac{n\;\times \;N\;\times \;Eq}{10\;\times \;p}$$

In which: *N* = normality of sodium hydroxide solution.

*n* = volume of sodium hydroxide solution spent in the titration in mL.

*p* = sample mass in grams = 1 g

*Eq*. = gram equivalent of acid = 64.02 (citric acid)

#### 2.2.2. pH

The pH of the pulp (25 °C) was determined in an Orion Expandable Ion Analyzer EA 940, pH meter [9].

#### 2.2.3. Soluble Solids

The concentration of soluble solids was determined with direct reading in a Carl ZEISS Jena bench refractometer, according to method no. 932.12 of the AOAC [9]. It was performed in triplicate at a temperature of 20 °C and the results were expressed in ° Brix.

### 2.3. Sensory Evaluations

#### 2.3.1. General Procedure

Sensory analyses were carried out in the Laboratory of Sensory Science and Consumer Studies of the Faculty of Food Engineering at UNICAMP in accordance with ISO 8589:2007 standards [10]. All tests were carried out in individual air-conditioned cabins (21 °C).

The blackberry nectar samples were prepared immediately before each test, as cited in the 2.1 item, to all sensory methods applied in the study.

The sequence of sensory methods applied was:(a)Ideal sweetness: determination of the concentration of sucrose to promote the ideal sweetness in blackberry nectar, realized by 120 consumers;(b)Selection and training of assessors to determine the equi-sweet of sweeteners (sucralose, aspartame, stevia RebA 78%, stevia RebA 92%, stevia RebA 97%, and tasteva) in the ideal sweetness determined to sucrose in blackberry juice; selection and training of assessors to time-intensity analysis (age 25–54 yo);(c)Sweetness equivalence: determination of concentration to equi-sweet of sucralose, aspartame, stevia RebA 78%, stevia RebA 92%, stevia RebA 97%, and tasteva in the same ideal sweetness determined to sucrose in blackberry juice carried out by 18 selected and trained assessors (age 25–54 yo);(d)Time-intensity analysis: The time-intensity analysis for each one of the descriptor terms (sweet taste, bitter taste, acidic taste, and blackberry flavor) was carried out with four repetitions by 18 selected and trained assessors (age 25–54 yo). The analysis of blackberry nectars with different sweeteners was studied in equi-sweetness to the ideal concentration of blackberry nectar with sucrose [11];(e)Acceptance analysis and check-all-that-apply (CATA): the acceptance and CATA were carried out by 116 consumers. They analyzed the samples in the equi-sweet of ideal sweetness to blackberry nectar with sucrose.

#### 2.3.2. Ideal Sweetness

First, a sensory analysis was performed with 120 consumers to determine the ideal sweetness using the ideal scale [12] to determine the ideal sucrose concentration in blackberry nectar. Five sucrose concentrations were presented (7%, 9%, 11%, 13%, and 15%). The ideal sweetness was determined through an unstructured 9 cm “Ideal Scale” that ranged from the left end as “extremely less sweet than ideal” to the right end as “extremely sweeter than ideal”, with the center of the scale being the ideal sweetness point. A value corresponding to −4.5 was defined at the extreme left point and a value corresponding to +4.5 was defined at the extreme right point, with the center of the scale being the point 0. The ideal sweetness result was determined through the linear regression of the data obtained in the test, according to the Vickers method [13]. The ideal concentration of sweetness predicts the optimum level of sucrose for the experiment, using the average of collected data with a nonstructured ideal scale correlated with each sucrose concentration used in the product studied.

The unstructured scale was chosen to provide consumers more freedom to express their sensory perception and assess with more acuity the decision point because it is possible to mark any point on the line scale [13,14].

#### 2.3.3. Assessors Selection and Training to Sweetness Equivalence and Time-Intensity Analysis

Tasters were preselected to participate in the tests to determine the equivalence of sweetness and for the time-intensity analysis with the objective of selecting candidates familiar with the product, having the ability to discriminate the samples, and having adequate skills to use the data collection program. Thus, a sequential Wald analysis was performed with 24 candidates, in which a series of triangular difference tests were presented using blackberry nectar sweetened with 4% and 5% sucrose (m/m). These concentrations were determined through a previous paired test with 30 tasters, where the two concentrations were presented, and it was identified that they differed at a significance level of 0.1%.

Wald’s sequential analysis presents some parameters for selecting these tasters, namely: *p*0, *p*1, *α* and *β*. One can define *p*0 as the expected proportion of correct decisions when the samples are identical, *p*1 as the expected proportion of correct decisions when the unequal sample is detected on half the total number of occasions, *α* as the probability of accepting a candidate without sensory acuity, and *β* as the probability of rejecting a candidate with sensory acuity [12,15].

Thus, 24 tasters were preselected, where the candidates were evaluated according to their performance in relation to two straight lines, expressed in Equations (2) and (3), constructed from the parameters *p*0, *p*1, *α* and *β*, which delimit acceptance, rejection, or continuation regions of the tests. The parameter values used in the test were: *p*0 = 0.33, *p*1 = 0.66, *α* = 0.05, and *β* = 0.10 [12].
(2)$$d0=\frac{\log\beta -\log\left(1-\alpha \right)-nlog\;\left(1-p1\right)+nlog\;\left(1-p0\right)}{\log p1-logp0-\log\left(1-p1\right)+\log\left(1-p0\right)}$$
(3)$$d1=\frac{\log(1-\beta )-\log\alpha -nlog\;\left(1-p1\right)+nlog\;\left(1-p0\right)}{\log p1-logp0-\log\left(1-p1\right)+\log\left(1-p0\right)}$$

Candidates were also selected for the time-intensity analysis based on experience, discrimination power, repeatability, and agreement with the team, being verified through two-factor analysis of variance (sample and repetition) for each panelist in relation to each parameter of the obtained curve [16]. The panelists with a significant F-sample (*p* < 0.30) and non-significant F-repetition (*p* > 0.05) and non-significant sample × taster interaction (*p* > 0.05) were selected for each parameter [16]. The panelists also participated in training, lasting approximately one h, to use the data collection and test simulation software program, in addition to the sensorial memorization of the references used for the maximum and minimum intensity of the evaluated stimuli. The training was applied for 10 h (two hours per day).

#### 2.3.4. Sweetness Equivalence

The sweetener concentrations that are equivalent to the same sweetness of the blackberry nectar sweetened with sucrose were determined using the magnitude estimation method [17]. The samples of blackberry nectars were presented as complete balanced blocks, together with a reference sample sweetened with sucrose at the concentration determined in the ideal sweetness test (9.3%), separately for sweetener [17,18,19]. The trained tasters received 5 samples in different concentrations (separately to each sweetener) at a temperature of 4 ± 2 °C and were asked to estimate the intensity of sweetness compared to the reference sample, which had an arbitrary sweetness value of 100 (the reference blackberry nectar with sucrose in ideal sweetness concentration). The concentrations used to each sweetener for composing the set of samples presented to the assessors are listed in Table 1. These concentrations were based on values determined by Correa and Bolini [19].

Normalization for data analysis was performed through the geometric mean of the estimated sweetness values for each sweetener and its respective concentration for each taster. The concentration scores were divided by the geometric mean of each taster, and the geometric means of each sample were calculated. Linear regression of log values of sweetener concentrations was performed. The equation used for linear regression is as follows:y = a + b·x(4)

a = y value at the intercept

b = slope of the line

The Power Function was used to determine the equivalent sweetness, represented in Equation (5):S = a·C ^n^
(5)

In which: S = Perceived sensation

C = Stimulus concentration

a = antilog of the y value at the intercept

e = Slope of the obtained line 

From this equation, it is possible to determine the perceived sensation (S) from the previously determined ideal sucrose concentration (C) in order to determine the ideal concentration of each sweetener.

#### 2.3.5. Time-Intensity Analysis

The samples of blackberry juice in the same equi-sweetness to the ideal predetermined, were presented monadically (30 mL) and sequentially at 14 ± 2 °C in 50 mL disposable cups coded with 3 random digits. The time-intensity analysis data were collected using the TIAFT (Time-Intensity Analysis of Flavors and Taste) software [11].

The descriptor terms evaluated in the time-intensity analysis were: sweet taste, bitter taste, acidic taste, and blackberry flavor. The attributes were evaluated individually, and the samples were evaluated monadically and in four repetitions, recording the intensity of the attribute as a function of the elapsed time on the monitoring scale using the mouse on a ten-point scale (0 = none; 5 = moderate; 10 = strong). The following parameters were provided by the program: maximum intensity; time when the maximum intensity was recorded; time after ingestion of the sample in which the evaluated attribute was no longer perceived by the taster; graph of the Time x Intensity curve and area under the Time x Intensity curve.

The maximum intensity reference was presented to the training of assessors and represented the maximum of the scale (10 = strong) and minimum (0 = none). Blackberry pulp with water in the proportion of 1:1 was considered for the reference of maximum blackberry flavor. The maximum intensity references of sweet taste, bitter taste, and acidic taste were prepared with blackberry pulp and water in the proportion of 1:2, added 15% of sucrose, 0.144% of stevia, 97% of rebaudioside A, and 9.3% of sucrose and 0.2% of citric acid, respectively. The minimum intensity for all descriptors was deionized water. The references were adapted from the method described by Medeiros and Bolini [3].

The data collected in the time-intensity analysis were evaluated by analysis of variance (ANOVA) and Tukey’s test, checking for differences between the samples with a significance level of 5% (*p* = 0.05) using the SAS software program (Statistical Analysis System, 2022, Raleigh, NC, USA).

#### 2.3.6. Acceptance Analysis

The acceptance analysis was performed with 116 consumers, representing the target public, 48% men and 52% women aged between 21 and 60 years. The acceptance in relation to appearance, presence of foam, aroma, flavor, texture, and overall impression was carried out using a nine-centimeter unstructured hedonic scale [18,19,20,21].

Consumers evaluated the seven samples of blackberry nectars with different sweeteners (sucrose, sucralose, aspartame, stevia Reb A 78%, stevia Reb A 92%, stevia Reb A 97%, and tasteva) in equi-sweetness. The samples were presented to consumers in a balanced complete block design in a sequential monadic way [3,18]

The purchase intention was analyzed by the consumers using a 5-point scale, rang-ing between “1 = would certainly not buy” and “5 = would certainly buy” [12]. The collected data were evaluated by analysis of variance (ANOVA) and Tukey’s test (*p* = 0.05) using the SAS software (Statistical Analysis System, 2022–Version 9.4, Raleigh, NC, USA).

Multivariate statistical analysis based on the principal component analysis was applied to the representation of individual notes of consumer acceptance in relation to overall impression to performing the internal preference map [22,23].

Data consumer acceptance in relation to overall impression also were correlated with the results of time-intensity analysis curves parameters using the PLSR (partial least squares regression), generating an external preference map using the XLSTAT software version 2022 (Addinsoft, Paris, France).

#### 2.3.7. CATA (Check-All-That-Apply)

In characterizing the samples using the CATA methodology, the same 116 consumers of acceptance analysis also were invited to mark all the terms that characterize the sample in a list of 16 descriptors terms, without marking limits [24], showed in the computer displayed questionnaire. A total of 16 descriptors were evaluated: sweetness, bitterness, residual bitterness, blackberry flavor, presence of foam, residual sweetness, homogeneous, heterogeneous, fluid, viscous, blackberry aroma, red color, acidic, full-bodied, shiny, and full-bodied.

The use of lists with fewer descriptor terms does not influence the characterization results [24]. The test was applied in sequence to the acceptance analysis with the 116 consumers. The terms were randomized, meaning that the terms appeared in random order among the tasters. Data analysis was performed using Cochran’s Q test, and correspondence analysis correlated with the overall impression data obtained in the acceptance analysis [24].

## 3. Results and Discussion

### 3.1. Physicochemical Characterization

The results found for the physicochemical analyses of the blackberry pulp are shown in Table 2.

There are pH values between 2.8 and 3.2, titratable acidity between 0.85% and 1.58%, and soluble solids between 5.37 and 11.1 Brix between blackberry variations in the literature [25,26,27,28]. The values may vary from study to study depending on the cultivar, climate, location, time of year, and maturation stage. However, it was found that there is a relationship between the concentration of soluble solids in the pulp and the rheological parameters. The greater the concentration of soluble solids, the greater the pulp viscosity. Apparent viscosity is also related to temperature, decreasing its viscosity when its temperature is higher [26].

### 3.2. Ideal Sweetness

The mean values of all the tasters for each sample were linearized as a function of the concentrations of each sample evaluated. It was possible to find the ideal sucrose concentration for the blackberry nectar from the equation of the obtained line, which is 9.32% (m/m). The value of 9.3% (m/m) was adopted to facilitate sample preparation (Figure 1).

### 3.3. Sweetness Equivalence

Figure 2 shows the relationship between sweetness intensity and sweetener concentration on a logarithmic scale. Sucrose, aspartame, and sucralose have a slope close to 1, which indicates linear behavior, meaning that the sweetness potency increases proportionally to the increase in the sweetener concentration. On the other hand, stevia presented a linear coefficient lower than 1, meaning that an increase in the sweetener concentration does not allow a proportional increase in the perception of sweetness. It was possible to calculate the equivalent concentration of sweetener through the potency function to obtain the same sweetness as sucrose. The equivalent concentrations are shown in Table 3.

Sucralose has a sweetness potency of 620 times compared to 9.3% sucrose in blackberry nectar. Aspartame has a sweetness potency of 180 times compared to 9.3% sucrose. Moreover, the sweetness potency for stevia with 78%, 92%, and 97% rebaudioside A is 96, 95, and 97 times, respectively. The potency equivalent to 9.3% sucrose for tasteva^®®^ brand stevia is 96.

Again, the sweetness potency of sucralose was found to be 620-fold with respect to 9.3% sucrose. In comparison, Medeiros [3] found a value of 509 for peach nectar sweetened with 8.6% sucrose. Freitas, Dutra, and Bolini [29] found a value of 625 for cherry nectar sweetened with 10% sucrose. Aspartame was found to have a sweetness potency of 180 times with respect to 9.3% sucrose. The same authors [29] found a value of 185 for cherry nectar sweetened with 10% sucrose. Finally, Correa and Bolini [19] found a value of 181 for a cashew drink sweetened with 9% sucrose.

### 3.4. Time-Intensity Analysis

In the results obtained through the time-intensity analysis (presented in Table 4), it was possible to verify that the samples with the highest intensity of sweet taste (Imax) were those sweetened with stevia and aspartame, not differing significantly between them (*p* ≤ 0.05). The sample sweetened with sucrose has a lower intensity of sweetness, not differing from the sucralose, aspartame, and stevia 78% and 97% of rebaudioside A samples. Different results are found in other products [3,19]. This result reinforces the necessity to apply the sensory studies specific to each product because the parameters of time intensity (for example, intensity, time of duration of stimulus) can differ in the function of the food matrix. The stevia samples present higher values than the others in evaluating the duration of sweet taste (Ttotal), not differing from the aspartame sample. Aspartame, sucralose, and sucrose did not differ significantly (*p* > 0.05) in relation to the total stimulus time.

The results obtained for the parameter area of the curve were concomitant with the others, in which the stevia samples had the highest value, while the sucrose and sucralose samples had the lowest values and did not differ significantly.

The samples that presented the highest stimulus intensity regarding the bitter taste (Imax) were those sweetened with stevia 97% and 92% of rebaudioside A. Tasteva stevia has a lower bitterness intensity than stevia 97 Reb A, not differing from stevia with 78 and 92% of rebaudioside A. Sucrose presented lower bitterness intensity, followed by sucralose and aspartame, which did not differ significantly (*p* ≤ 0.05).

The samples that presented the longest stimulus time for the duration of bitterness (Ttotal) were the stevia samples with 78%, 92%, and 97% of rebaudioside A, which did not differ from each other. Sucrose presented the shortest time, not differing from the sucralose and aspartame samples. The samples with the highest values for the area under the curve were stevia with 92% and 97% rebaudioside A, which did not differ significantly from each other. Tasteva presented an intermediate value, not differing from the samples of stevia with 78% and 97% of rebaudioside A. Sucrose, aspartame and sucralose presented lower values without statistically differing (*p* < 0.05) between them.

The only parameter that showed a significant difference (*p* ≤ 0.05) regarding sour taste in at least one sample was the time to reach maximum intensity (Imax). The stevia sample with 78% of rebaudioside A has a longer time, while sucralose has a shorter time. No attribute showed a significant difference (*p* ≤ 0.05) regarding blackberry flavor (*p* > 0.05).

### 3.5. Acceptance Analysis

The acceptance in relation to appearance, presence of foam, aroma, flavor, texture, and overall impression was evaluated by consumers. The average scores given by consumers for the evaluated attributes are shown in Table 5.

The sample sweetened with aspartame presented the highest average in the evaluation of appearance, differing only from the sample sweetened with sucrose. Regarding the presence of foam, there was no significant difference between the samples (*p* ˃ 0.05).

The sample sweetened with sucralose had the highest acceptance average regarding the aroma, not differing from the sucralose, stevia 92%, stevia 78% of rebaudioside A, tasteva, and sucrose samples (*p* > 0.05).

The sample sweetened with aspartame presented the highest average in the evaluation of appearance, differing significantly only from the sample sweetened with sucrose. Regarding the presence of foam, there was no significant difference between the samples (*p* > 0.05).

The blackberry nectar sweetened with aspartame and stevia 92%, 78%, and 97% of rebaudioside A had the lowest average acceptance regarding flavor, not differing from each other at the significance level of 5%.

The blackberry nectar sweetened with sucralose had suitable acceptance, not differing from the sucrose and tasteva samples (*p* > 0.05).

The sucrose, sucralose, and tasteva promote the samples with higher acceptance regarding flavor and overall impression, not differing from each other (*p* < 0.05). The samples sweetened with stevia had a lower average acceptance; however, they did not differ from the sucralose, sucrose, and tasteva samples in relation to appearance and aroma. Moreover, sucralose and tasteva had greater acceptance regarding the flavor and overall impression. Samples sweetened with stevia did not differ from each other and had a lower average overall acceptance.

According to the purchase intent histogram of the samples (Figure 3), the product that had the highest positive purchase intent (would certainly buy and would probably buy) was the sucrose sample (67.08%), followed by the aspartame and sucralose also had high positive purchase intent percentages of 58.48% and 48.16%, respectively. This result can be explained by the presence of lower intensity of bitter taste and total bitter taste duration in these samples. As the stevia tasteva had a shorter bitter taste duration, but it obtained a positive purchase intent with the same tendency compared to the other samples sweetened with stevia (tasteva = 30.96%; stevia RebA 78 = 31.82%; stevia Reb A 92 = 29.24%; stevia Reb 97 = 22.36%).

The results were also analyzed through the internal preference map (Figure 4), in which points are positioned in the multidimensional space referring to acceptance in relation to the overall impression of consumers.

The internal preference map was obtained by principal components 1 and 2, which together explained 49.52% of the variation between samples. This value is relatively low; however, it is considered adequate for consumer acceptance results, as the liking values can vary in a large range, which also needs to be measured in this study [12]. Other published studies have shown concordant results in other products [3,19]. The dispersion of consumers’ acceptance of individual notes to overall liking (red dot points) is evidenced in Figure 4 because the consumers’ preference spread near all samples (blue diamonds). This preference is evidenced by the proximity of the red dot by a determined sample. As it is possible to observe that the taster closer to the nectar with sucrose is the nectar sweetened with sucralose, which indicates a greater preference for these samples. These samples obtained higher acceptance averages by performing Tukey. On the other hand, there is little concentration of tasters around the stevia with 78%, 92%, and 97% rebaudioside A and tasteva samples, which indicates a lower preference of consumers for these samples, and which can be proven through the overall impression analysis of the Tukey test. It is important to observe the purchase intention results that are in concordance with the preference data. These results evidence the importance of the application of sensory analysis to each specific product because the results can vary drastically in function to the food matrix. To blackberry nectar, the results were interesting because it presents values different from other products.

### 3.6. Preference Temporal Drivers Generated by Partial Least Square Regression (PLSR)

Time intensity and overall liking data were analyzed by multivariate statistics of partial least squares regression (PLSR), thus generating the temporal preference drivers for blackberry nectar.

Figure 5 shows the parameters that positively contribute to the acceptance of blackberry nectar, which are the maximum intensity of blackberry flavor, total blackberry flavor stimulus time, and blackberry flavor stimulus area. They present significant and positive coefficients. The parameters that negatively contribute to the acceptance of the samples are total sweet taste time, sweet stimulus area, maximum bitter taste intensity, maximum bitter taste intensity time, total bitter taste time, and bitter stimulus area. The parameters maximum intensity of sweet taste, maximum intensity of sweet taste time, maximum intensity of blackberry flavor time, maximum intensity of acidic taste, maximum intensity of acidic taste time, total acidic taste time, and acid stimulus area did not show a suitable correlation, and it was not possible to say whether they positively or negatively contributed to the acceptance of blackberry nectar.

Thus, greater acceptance by consumers is observed in samples correlated with higher temporal parameter values of the blackberry flavor, thus denoting the importance of the sweetener in not only promoting the sweet taste but also with the least possible interference in the perception of the blackberry flavor. A negative influence on the acceptance of the samples can also be noted for those that presented higher values in the parameters related to bitter taste, which demonstrates the importance of using sweeteners that intensify the perception of bitter taste as little as possible. It is also observed that despite the intensity of sweet taste not negatively contributing to the acceptance, a very pronounced total time of sweetness negatively contributes to acceptance, as seen in the parameter total sweetness time as a negative coefficient. This reinforces the importance of choosing a sweetener with the lowest possible sweet residual taste.

A sample characterization test was carried out through consumer perception using the CATA (Check-All-That-Apply) test. This test is statistically analyzed using Cochran’s Q test. A low p-value indicates that the products differ significantly from each other. For the analysis herein, the attributes that were not significant are: homogeneous, heterogeneous, red, opaque, and glossy. The quality of the analysis is suitable, as it had an 87.42% of explanation for the first two dimensions.

Figure 6 represents the correspondence analysis of the data obtained from CATA. The samples are located next to the words that characterize them. The blackberry nectars sweetened with aspartame and sucrose are characterized by having the attributes of blackberry flavor, blackberry aroma, and sweetness. The nectar sweetened with sucralose is characterized by full-bodied, viscous, and the presence of foam.

The samples with stevia 97% RebA and 92% RebA are close together in the multidimensional space and present bitter and residual bitter attributes. Stevia 78% RebA and tasteva samples are characterized by presenting a sweet residual taste and being fluid and acidic.

C.d.M.a shows the principal coordinate analysis results, and Figure 7b the mean impact of descriptors from CATA to blackberry nectar. It is possible to verify that the descriptors localized near the overall impression are important and can contribute positively to acceptance. Based on this information, the color red, sweet, blackberry flavor, blackberry flavor, residual sweet, and glossy are important to the consumer in this product. Figure 7b represents the impact of the descriptor on consumer acceptance. The size of the bar, starting from the vertical line (zero), is proportional to the importance of the respective attribute. The greater the length of the bar, the greater the importance of the attribute in the average of consumers’ acceptance in relation to the overall impression.

It can be concluded that the attributes that the samples must present to be well accepted, as observed through the overall impression, are blackberry aroma, blackberry flavor, sweet taste, and glossy.

Figure 7b indicates the attributes that positively contribute (blue bars) and those that negatively contribute (red bar) to the acceptance of the samples. The blackberry flavor promotes a higher positive impact on preference in blackberry nectar indicated to be present in the sample, and the residual bitterness is indicated not to be present in the sample.

The blackberry flavor has the greatest positive impact on the increasing preference for blackberry nectar, followed by other descriptors indicated in the blue bars in Figure 7b. That is, the greater the presence of blackberry flavor in the nectar, the greater its acceptance. Therefore, it is an important sensory characteristic for consumers, and it needs to be present in this nectar. Conversely, the red bar indicates that residual bitterness should not be present in blackberry nectar because it is an attribute that causes a negative impact on the overall impression mean.

## 4. Conclusions

The results obtained in this study evidenced the important role of sensory science and consumer studies in finding the decision to choose the better formulations in food studies, applying different methods according to the specific objectives. Likewise, the magnitude estimation method is an important method to determine the equi-sweet concentration of sweetener relative to sucrose, specific for different matrices. Therefore, validation of sensory testing with assessors and product consumers is necessary to determine the ideal formulation, as well as to obtain the highest acceptance.

The study of the dynamic sensory profile of beverages sweetened with sweeteners enables us to understand the intensity of characteristics of foods during consumption time. In turn, it is possible to determine which sweetener most closely matches the sucrose profile in different products. According to this study, the better sweetener to replace sucrose in blackberry nectar is aspartame and sucralose.

The drivers of preference to blackberry nectar were the time-intensity parameters of maximum intensity of blackberry flavor, total blackberry flavor time, and the blackberry flavor stimulus area.

The CATA (Check-All-That-Apply) characterization test is a quick test performed by consumers. From this test, it was possible to demonstrate that the words of sensory characteristics related to suitable acceptance of the overall impression were blackberry aroma, blackberry flavor, sweet taste, and glossy. The samples that were most characterized by these attributes were the samples sweetened with aspartame and sucrose.

## Figures and Tables

**Figure 1 foods-12-00549-f001:**
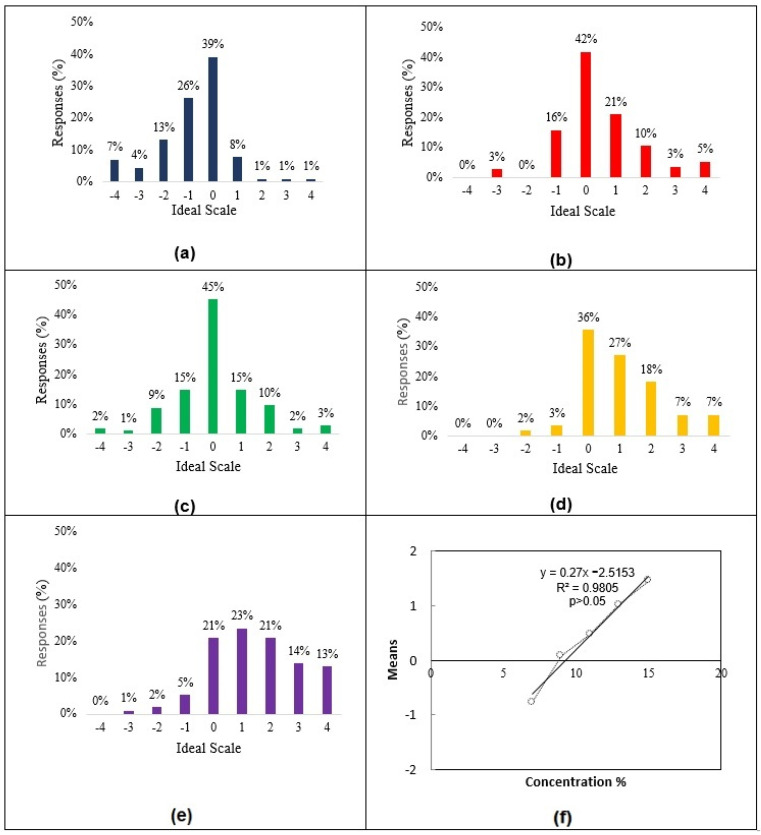
Histogram of the responses for the Ideal Sweetness test, with different sucrose concentrations: 7.0% (**a**) blue bars, 9.0% (**b**) red bars, 11.0% (**c**) green bars, 13% (**d**) yellow bars, and 15% (**e**) purple bars, and straight-line graph and equation obtained from linearizing the Ideal Sweetness test results (**f**).

**Figure 2 foods-12-00549-f002:**
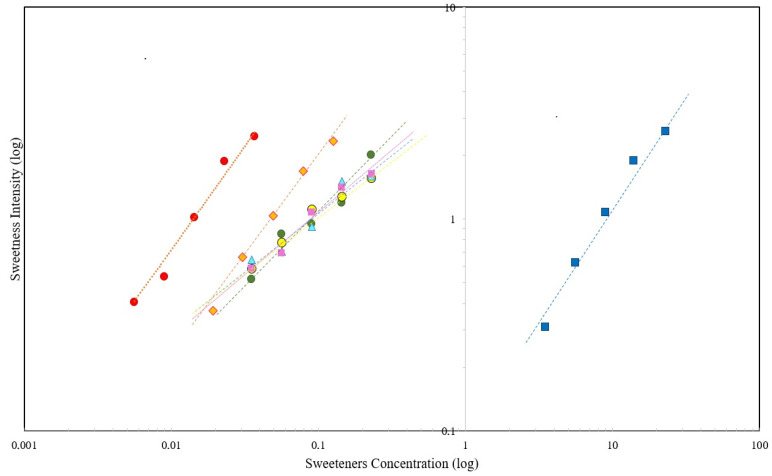
Concentration and sweet stimulus intensity on a logarithmic scale.

**Figure 3 foods-12-00549-f003:**
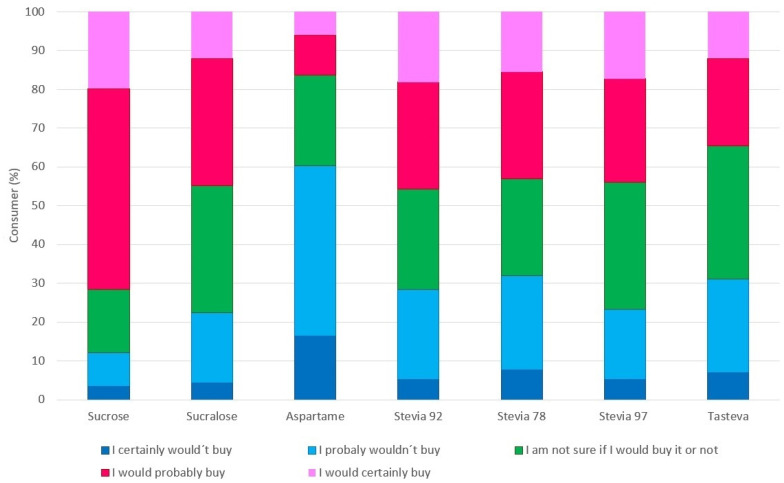
Histogram of purchase intention for blackberry nectar samples.

**Figure 4 foods-12-00549-f004:**
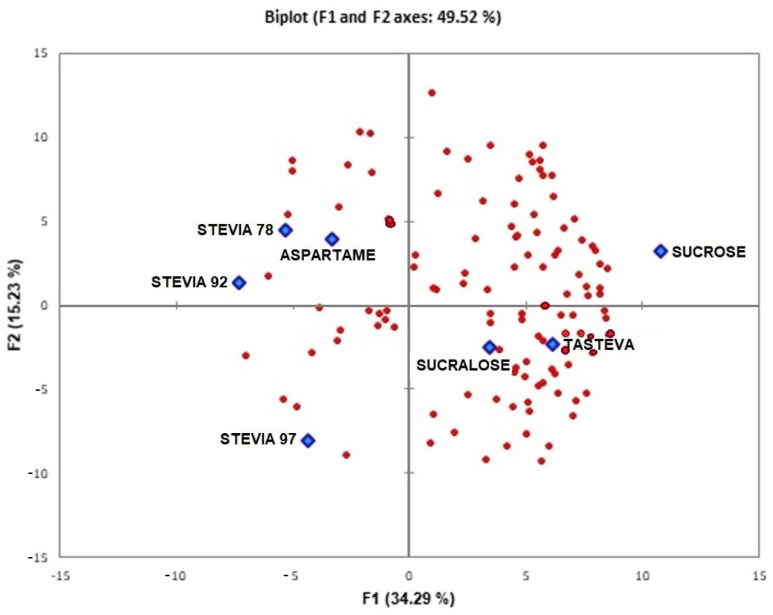
Internal preference map of blackberry nectar referring to the overall impression (Red dots are consumers and blue diamonds are sweeteners used in blueberry nectar).

**Figure 5 foods-12-00549-f005:**
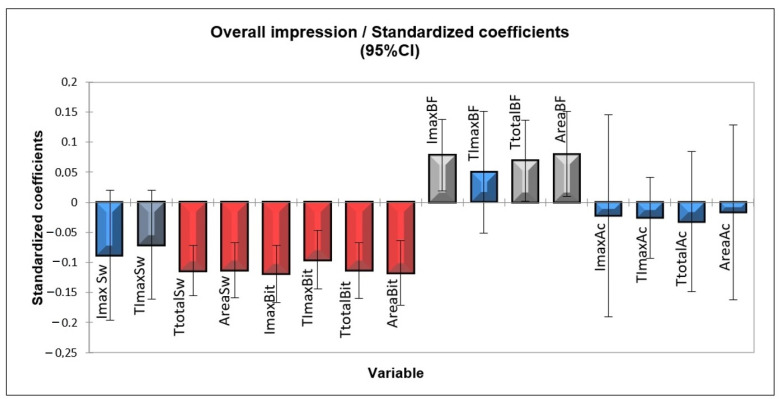
Preference drivers of blackberry nectar. Legend: ImaxSw = Maximum intensity of sweet taste; TImaxSw = Maximum intensity time to sweet taste; TtotalSw = Total time of sweet taste; AreaSw = Area under the curve of sweet taste stimulus; ImaxBit = Maximum intensity of bitter taste; TImaxBit = Maximum time to intensity of bitter taste; TtotalBit = Total time to bitter taste; AreaBit = Area under the curve of bitter taste stimulus; ImaxBF = Maximum intensity of blackberry flavor; TImaxBF = Maximum time to intensity of blackberry flavor; TtotalBF = Total time blackberry flavor; AreaBF = Area under curve of blackberry flavor stimulus; ImaxAc = Maximum intensity of acidic taste; TImaxAc = maximum intensity time to of acidic taste; TtotalAc = Total time acidic taste; AreaAc = Area under the curve of acidic taste stimulus. The PLSR analysis graphically shows the positive or negative importance of the parameter’s time-intensity curves, evidencing the consumers’ preference drivers. The columns on the positive part of the y-axis (gray) represent parameters with positive importance, whereas the columns on the negative part of the y-axis (red) represent parameters whose presence is negative to acceptance of the blackberry nectar. The vertical lines represented in the boxes (columns) of the parameters of time-intensity curves (coefficients) are the confidence intervals. If the confidence interval exceeds (crosses) the x-axis to the opposite side, it means that the corresponding parameter is not significant (*p* > 0.05) in the consumer’s preference (blue columns). However, if the confidence interval does not cross the x-axis to the opposite side, this coefficient (parameter of the curve) is significant for consumer preference (*p* < 0.05), and it contributes positively or negatively to consumer preference if the bar is facing the positive or negative side of the y-axis, respectively.4. CATA (Check-All-That-Apply).

**Figure 6 foods-12-00549-f006:**
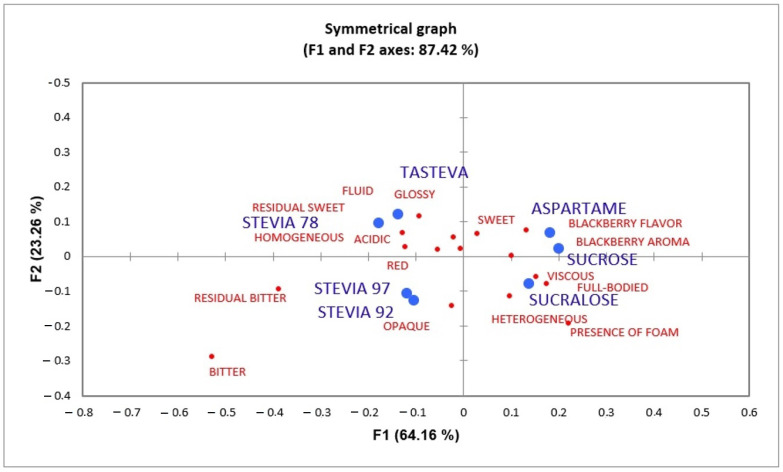
Correspondence analysis of “check-all-that-apply” (CATA) words. Red dots are words used in the CATA and blue dots are the sweeteners applied in the blackberry nectar.

**Figure 7 foods-12-00549-f007:**
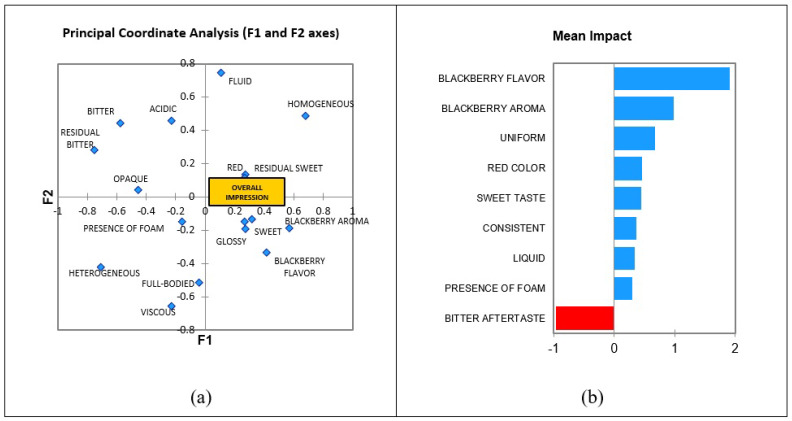
Principal coordinate analysis (**a**) and mean impact of blackberry nectar (**b**). Blue diamonds are words used in the CATA.

**Table 1 foods-12-00549-t001:** Sweeteners concentration to determine sweetness equivalence.

Sweeteners	Concentration (%)				
Sucrose	3.5200	5.6300	9.0000	14.0700	23.0400
Sucralose	0.0056	0.0090	0.0144	0.0230	0.0369
Aspartame	0.0193	0.0306	0.0495	0.0792	0.1267
Stevia 78 RebA	0.0352	0.0563	0.09	0.144	0.2304
Stevia 92 RebA	0.0352	0.0563	0.09	0.144	0.2304
Stevia 97 RebA	0.0352	0.0563	0.09	0.144	0.2304
Tasteva	0.0352	0.0563	0.09	0.144	0.2304

**Table 2 foods-12-00549-t002:** Physicochemical characterization of blackberry pulp *.

Characterization	Results
Total titratable acidity	1.21 ± 0.01 **
pH	2.93 ± 0.02
Total soluble solid	10.25 ± 0.10 ***

* Values expressed as the mean ± standard deviation; ** %citric acid; *** Brix).

**Table 3 foods-12-00549-t003:** Concentration equivalent to 9.3% of sucrose and equivalent power of sweetener in blackberry nectar.

Sweeteners	Concentration Equivalent to 9.3% of Sucrose (%)	Potency Equivalent to 9.3% of Sucrose In Blackberry Nectar
Sucralose	0.015	620
Aspartame	0.052	180
Stevia 78 RebA	0.097	96
Stevia 92 RebA	0.098	95
Stevia 97 RebA	0.096	97
Tasteva	0.097	96

**Table 4 foods-12-00549-t004:** Means * of the parameters from time-intensity curves for the sweet taste, bitter taste, acidity, and blackberry flavor to the blackberry nectar in the equi-sweet ideal concentration.

Parameters Curves	Sucrose	Sucralose	Aspartame	Stevia 92	Stevia 78	Stevia 97	Tasteva
**Sweet Taste**							
**Imax**	7.17 ^c^	7.50 ^b,c^	7.8 7 ^a,b,c^	8.25 ^a^	7.71 ^a,b,c^	7.77 ^a,b,c^	8.03 ^a,b^
**TImax**	7.34 ^a^	8.34 ^a^	7.49 ^a^	8.074 ^a^	8.66 ^a^	8.48 ^a^	7.49 ^a^
**Ttotal**	37.38 ^b^	39.88 ^b^	46.83,^a,b^	60.44 ^a^	59.82 ^a^	60.08 ^a^	60.43 ^a^
**Area**	127.89 ^c^	150.73 ^c^	175.21 ^b,c^	224.15 ^a,b^	230.23 ^a,b^	240.58 ^a^	251.82 ^a^
**Bitter Taste**							
**Imax**	3.17 ^e^	4.32 ^d^	4.35 ^d^	7.51 ^a,b^	6.33 ^c^	7.91 ^a^	6.57 ^b,c^
**TImax**	6.04 ^a^	5.71 ^a^	6.27 ^a^	7.42 ^a^	7.26^a^	8.51 ^a^	6.69 ^a^
**Ttotal**	16.71 ^c^	22.89 ^c^	24.99 ^b,c^	45.85 ^a^	45.18 ^a^	46.28 ^a^	33.38 ^b^
**Area**	27.82 ^c^	58.01 ^c^	64.10 ^c^	187.20 ^a^	133.73 ^b^	175.73 ^a,b^	130.73 ^b^
**Acidity**							
**Imax**	5.40 ^a^	5.73 ^a^	5.85 ^a^	5.82 ^a^	5.69 ^a^	5.46 ^a^	5.70 ^a^
**TImax**	5.89 ^a,b^	4.93 ^b^	5.94 ^a,b^	5.74 ^a,b^	6.83 ^a^	5.81 ^a,b^	5.92 ^a,b^
**Ttotal**	19.67 ^a^	20.97 ^a^	20.09 ^a^	20.98 ^a^	19.81 ^a^	20.04 ^a^	20.08 ^a^
**Area**	60.88 ^a^	72.16 ^a^	68.38 ^a^	71.68 ^a^	64.40 ^a^	61.31 ^a^	67.60 ^a^
**Blackberry Flavor**							
**Imax**	7.26 ^a^	7.33 ^a^	6.72 ^a^	6.83 ^a^	6.85 ^a^	6.49 ^a^	6.48 ^a^
**TImax**	6.80 ^a^	6.47 ^a^	7.40 ^a^	6.24 ^a^	6.38 ^a^	6.32 ^a^	7.43 ^a^
**Ttotal**	20.54 ^a^	20.38 ^a^	19.30 ^a^	19.85 ^a^	18.27 ^a^	18.06 ^a^	19.63 ^a^
**Area**	72.33 ^a^	69.67 ^a^	66.37 ^a^	69.34 ^a^	62.78 ^a^	58.11 ^a^	57.36 ^a^

* Means with the same letters on the same raw do not differ statistically by Tukey’s test (*p* ≤ 0.05).

**Table 5 foods-12-00549-t005:** Means * of acceptance in relation to appearance, foam, aroma, flavor, texture, and overall impression of blackberry nectar to the blackberry nectar in the equi-sweet ideal concentration.

Acceptance	Sucrose	Sucralose	Aspartame	Stevia 92	Stevia 78	Stevia 97	Tasteva
Appearance	6.20 ^b^	6.48 ^a,b^	6.81 ^a^	6.53 a,b	6.51 ^a,b^	6.54 ^a,b^	6.49 ^a,b^
Foam	5.77 ^a^	5.95 ^a^	5.97 ^a^	5.90 ^a^	5.88 ^a^	5.94 ^a^	5.81 ^a^
Aroma	5.54 ^a^	5.97 ^a^	5.34 ^b^	5.51 ^a,b^	5.49 ^a,b^	5.40 ^b^	5.78 ^a,b^
Flavor	5.37 ^b^	5.87 ^a,b^	4.48 ^c^	4.30 ^c^	4.40 ^c^	4.46 ^c^	6.38 ^a^
Texture	6.19 ^a^	6.37 ^a,b^	5.95 ^b^	5.92 ^b^	5.96 ^b^	6.03 ^b^	6.67 ^a^
Overall impression	5.71 ^b^	6.05 ^a,b^	5.13 ^c^	4.89 ^c^	4.92 ^c^	5.05 ^c^	6.31 ^a^

* Means with the same letters on the same raw do not differ statistically by Tukey’s test (*p* ≤ 0.05).

## Data Availability

The authors declare that all the data supporting the findings of this study are available within the article.

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
