# Peer review of "Preference Drivers for Blackberry Nectar (Rubus spp., Rosaceae) with Different Sweeteners"

_foods, 2023, doi:10.3390/foods12030549_

Round 1

Reviewer 1 Report

The author uses a large number of sensory methods to determine the sweet effects of different sweeteners, which is a good idea. However, the following major issues should be addressed before the manuscript is published in the Foods.

1.Just-about-right and PLSR were mentioned in the abstract, but the corresponding content were not found in the method and discussion in the manuscript.

2. L315-L346, “Acceptance analysis” The language expression is not good, and there are many inconsistencies with the data in Table 5, such as L322 and L328.

3.Figure 4 needs to be annotated as to what the red dot represents.

4. L362, This value is relatively low and can be explained by the fact that consumers’ preference is divided between more than one sample. What's the evidence for that?

Reviewer 2 Report

Review Foods-2089547

The article contains interesting research, but needs to be completed and rewritten in order to be published. Below are my proposals, in order to refine the manuscript:

Line 2

The title should be changed. If you add sweetener to a juice it is no longer a juice it can be called a nectar, a beverage, etc… It can be beverage sweetened by different sweeteners.

The same comments refer to the whole text in the manuscript when you call the product juice. Please correct it.

Line 9

The abstract need correction. Some statements do not correspond to the data in the study results.

Line 30

Perhaps it is worth mentioning in the introduction the sweeteners that were selected for the study and why they were chosen.

Line 71 Material and methods

A number of sensory methods were used in the study, which is a great asset, but there is little explanation of the individual methods, the rationale and purpose of their use, the way the study was conducted, the selection of samples to be evaluated and the assessors. What would have made the work clearer would have been to draw up an outline of the tests, including which were the first to be carried out, which samples were evaluated and who exactly evaluated them.

Line 105

There is a lack of information on how the juice was prepared, how it was diluted and how the beverage was prepared based on sweeteners and concentrate to use in whole 4 methods.

There is lack of tables with the formula of beverages used in whole 4 methods. It should be added. There is only table 1 with sweeteners concentrations.

Line 110

You have used TI, CATA, just about right scale, and magnitude estimation. Did you use samples prepared in the same way for all the methods? It should be specified.

Please explain which methods was used by consumers and which by trained pannelist?

Line 116

Did you use ideal scale or just about right scale?

Line 130

From the text we can deduce that only one test was used to train the panellists. this needs to be clarified, it is about how they were trained. Especially with the time intensity method, expert experience is extremely important.

Information on their age and gender should also be given.

Line 147

Especially with the time intensity method, expert experience is extremely important. Did they have it?

Line 166 table 1

How the sweetness of the sweeteners was evaluated?

Is it sweetness evaluated based on water solutions or in some juices, beverages etc?

Line186

Did you evaluate control sample of the juice without sweeteners, to know if the bitterness is not from the juice ?

Line 187

Please add reference to Time intensity method?

Line 201

It is not clear? Please explain the formula of beverages. Was the citric acid added in the beverages?

Why the QDP analysis was not used for the profile of beverages analysis?

Line 224

CATA method should be explained with the references cited.

Line 235

In whole manuscript the discussion is missing. The results should be discussed.

Line 253

Figure 1 e. Based on the results, Is the 15% concentration the proper one?    

Line 283

The results should be discussed. The results about the durability of the sucralose are different from those indicated in the literature.

Line 312 and 325

Table 4  and table 5    what about results with control sample with only juice?   

Line 316-346

The description of the results does not agree with the study results given in Table 5, this should be corrected .

Line 347

Please put the samples in the same order like in the previous research.

Line 351

Based on the results only aspartame purchase decision was so high like sucrose. Interpretation of the results should be changed and write based on the obtained data.

Line 424

Lack of explanation of the figure 7a, and principal coordinate analysis.

Line 430 and 464

I did not see in the methodology “brightness” among the evacuated attributes. Please correct it.

Line 452

Sweeteners instead of edulcorates

Line 455-456

Please write the conclusion based on the results, and achieved data.

Line 461

There is no comparison of QDP with CATA in this paper so it cannot be stated in this paper                                                                                                                                                                      

Line 484

Please prepare the list of references according to the journal requirements

Reviewer 3 Report

The Authors have chosen an important topic. In product development and in food science there is a constant need to balance between the sweetness of products and the optimal energy intake. Several studies have shown, that changing the sweetener composition may alter other sensory attributes, so this is an area where there are still several questions are open.

The chosen methodologies (JAR scale, magnitude estimation method, sensory profiling, time intensity analyses) are showing that the Authors aimed to have a holistic view in their study. That’s a strength of the paper.

Writing a good abstract is challenging, so I understand that the Authors wanted to emphasize the most important outcomes. However, we have to consider that blackberry is a plant, which has many varieties, and its quality greatly varies with the agrotechnical issues (e.g. fertilization, irrigation, etc) and the climatic and soil conditions. So I recommend that the following section, have to be re-phrased:

Lines 13-15 The optimal sucrose concentration in blackberry juice is 9.3% and the sweetener concentrations equivalent to sucrose was 0.015% of sucralose, 0.052% of aspartame and 0.09% of Stevia with different Rebaudioside A concentrations.

From the current text the reader might have the false impression, that if a blackberry juicy should be created, than 9.3% sucrose concentration will be always optimal. However that’s not the case. The optimal level of sweetness strongly depends on the group of consumers who were involved in the study. So I recommend to make a re-wording, which refers something like that: “According to our results and the opinion of the involved consumers….etc). Findings are important things in science, but they have to be treated properly.

Line 110: The samples were presented monadically (30 ml) and sequentially at 4 ± 2ºC

In consumer studies we usually serve the samples as they are expected to be consumed generally, in that case 4 Celsius might be sensible. But for profiling studies we usually apply higher temperatures, in order of better taste and flavor perception and discrimination. Please explain here briefly, why you used such a low temperature.

Lines 118-121: The ideal sweetness was determined through an unstructured 9 cm scale that ranged from the left end as “extremely less sweet than ideal” to the right end as “extremely sweeter than ideal”, with the center of the scale being the ideal sweetness point.

This description is really similar to the JAR scale. Although JAR structure can be applied to any type of sacles (in this case unstructured line scale), in many papers we find it with category scale approach. Please explain briefly, why did you chose unstructured scale instead of category scale.

In Table 4 and Table 5 the heading has too large font size, it should be reduced, for better layout, currently the names of the first 3 sweeteners are broken in several lines.

Since the JAR scale and the ideal scale were mentioned in the manuscript, I was expecting the use of Penalty Analysis (which is usually the evaluation method of JAR data). However this is not in the study. Please explain briefly, why did you use a different method of statistics.

Line 452 T’he study of the dynamic sensory profile of beverages sweetened with edulcorates,’

This word (edulcorate) is very rare in English, please use a different expression.

Line 461: e to descriptive methods performed with trained advisors.

Instead of ‘advisors’ please use ‘assessors’.

Round 2

Reviewer 1 Report

It has been modified as required.

Author Response

Author's Reply to the Review Report (Reviewer 1)

Point 1: English language and style are fine/minor spell check required

 Response 1: Dear Reviewer: We are thankful for all your comments and corrections, which contributed to the improvement of the manuscript. We accepted all of them, with gratitude.

As suggested, the corrected version of the manuscript was checked by a native English-speaking.

Thank you

The authors

Reviewer 2 Report

Thank you for correcting the text as recommended, which definitely improved the readability and clarity of the article. 

Author Response

Dear Reviewer: We are thankful for all your comments and corrections, which contributed to the improvement of the manuscript. We accepted all of them with gratitude.
As an author's Reply to the Review Report we inform grateful that the introduction and results were improved with the insertion of information, also, the corrected version of the manuscript was checked by a native English-speaking.
